# Computational Study of Network and Type-I Functional Divergence in Alcohol Dehydrogenase Enzymes Across Species Using Molecular Dynamics Simulation

**DOI:** 10.3390/biom14111473

**Published:** 2024-11-19

**Authors:** Suhyun Park, Petrina Jebamani, Yeon Gyo Seo, Sangwook Wu

**Affiliations:** 1Department of Physics, Pukyong National University, Busan 48513, Republic of Korea; suhyun@pukyong.ac.kr (S.P.); yeongs0709@pukyong.ac.kr (Y.G.S.); 2Department of Chemical Engineering, Pusan National University, Busan 46241, Republic of Korea; petrina@pusan.ac.kr; 3PharmCADD, 1102-ho, 60, Centum Jungang-ro, Haeundae-gu, Busan 48059, Republic of Korea

**Keywords:** alcohol dehydrogenases (ADH), network theory, molecular dynamics simulation, type-I functional divergence, posterior probability

## Abstract

Alcohol dehydrogenases (ADHs) are critical enzymes involved in the oxidation of alcohols, contributing to various metabolic pathways across organisms. This study investigates type I functional divergence within three ADH1 families: *Saccharomyces cerevisiae* (PDB ID: 4W6Z), *Gadus morhua* (PDB ID: 1CDO), and *Homo sapiens* (PDB ID: 1HDX). Understanding the molecular evolution and mechanisms underlying functional divergence of ADHs is essential for comprehending their adaptive significance. For this purpose, we performed a computational analysis that included structural characterization of ADHs through three-dimensional modeling, site-specific analysis to evaluate selective pressures and evolutionary constraints, and network analysis to elucidate relationships between structural features and functional divergence. Our findings indicate substantial variations in evolutionary and structural adaptations among the ADH families.

## 1. Introduction

Alcohol dehydrogenases (ADH) are a group of enzymes that have attracted significant scientific interest due to their fundamental role in the metabolism of alcohol across a diverse range of organisms, including bacteria, algae, yeasts, plants, and vertebrates [1,2]. These enzymes catalyze the transformation of alcohol into aldehyde or ketone with the concurrent reduction of nicotinamide adenine dinucleotide (NAD+) (Figure 1a). While this reaction yields potentially toxic intermediates, it also generates aldehyde and alcohol groups that serve various physiological functions [3]. ADH enzymes are particularly vital in yeasts, plants, and many bacterial species, where they drive fermentation processes critical for replenishing NAD+ (Figure 1b,c) and sustaining key metabolic pathways, thereby supporting survival and growth in low-oxygen environments [2,4]. In addition to their metabolic roles, ADH enzymes are a subject of evolutionary study due to their involvement in local adaptation, a process that promotes genetic variation and functional specialization across organisms. Investigating these adaptations provides valuable insights into how ADH genes evolve and acquire distinct roles relevant to organismal health and environmental adaptation.

Alcohol dehydrogenase (ADH) enzymes form a diverse gene family with several distinct classes. In yeasts and bacteria, ADH enzymes—typically functioning as tetramers with two zinc ions per monomer—catalyze reactions crucial for energy production through the conversion of acetaldehyde to ethanol. These functions are particularly important under conditions of low oxygen or low temperatures, enabling these organisms to adapt to environmental stressors [2,3]. ADH enzymes in plants similarly contribute to the maintenance of cellular redox balance and NAD+ regeneration [4,5]. In humans, ADH enzymes are structurally diverse, with at least seven unique genes encoding different isoforms. Human ADH is dimeric and incorporates two zinc ions (Zn^2+^), reflecting its intricate molecular architecture [6]. Among these, Class I ADHs (ADH1A, ADH1B, ADH1C) are liver-specific and central to ethanol metabolism [7], while Class II (ADH4), Class III (ADH5), and Class IV (ADH7) enzymes differ in substrate specificity and tissue distribution, highlighting the diversity within the human ADH family. Class II ADHs (ADH4) are also liver-based but are less efficient at metabolizing ethanol at low concentrations. Class III ADHs (ADH5) are found in nearly all tissues and exhibit broad substrate specificity. Class IV ADHs (ADH7), located in the stomach and esophagus, are effective at metabolizing other alcohols, such as retinol [8,9,10].

Gene duplication plays a key role in the evolution and diversification of ADH enzymes, as duplicated genes can evolve at different rates, a process known as type-I functional divergence. This divergence allows for the development of novel functions that enable organisms to adapt to their environments [11]. Function changes with evolution through processes like gene duplication, mutation, and selection, which allow proteins to develop new roles. For example, in the study by Carrijo de Oliveira et al., an ancestral enzyme (HIUase) involved in uric acid breakdown was duplicated, creating a “spare” gene copy [12]. Mutations in this duplicate allowed it to gradually lose its original enzymatic function and adopt a new role as transthyretin (TTR), a thyroid-hormone-binding protein. This transformation illustrates how structural modifications after duplication can lead to significant shifts in protein function, providing new biological capabilities over evolutionary time.

Thompson et al. explored the evolutionary functional diversification in plant ADHs and developed the first three-dimensional model of a plant ADH. Their study revealed site-specific amino acid changes in three crucial regions: the zinc-binding loop, the monomer interaction site, and the active site [13]. They also examined functional divergence in ADH enzymes across *Poaceae*, *Brassicaceae*, *Fabaceae*, and *Pinaceae* [14] identifying divergent amino acids in three critical regions: the zinc-binding loop, the monomer interaction site, and the active site. Additionally, they investigated ADH classes in animals, fungi, and plants, using evolutionary and modeling techniques to pinpoint key residues involved in different types of functional divergence between duplicated genes, providing insights into the process of ADH diversification [15].

The complex interplay between molecular evolution, structural diversity, and functional adaptation makes ADH enzymes an interesting subject for computational analysis. Techniques such as molecular dynamics simulations, integrated with network analysis, provide a powerful framework to investigate these relationships [16].

In this study, we investigate the molecular evolution (type-I functional divergence) of three ADH1 families (*Saccharomyces cerevisiae* (PDB id: 4W6Z, yeast alcohol dehydrogenase I) [17], *Gadus morhua* (PDB ID: 1CDO, cod(fish) alcohol dehydrogenase I) [18], and *Homo sapiens* (PDB ID: 1HDX, human alcohol dehydrogenase I) [19]) using network analysis and functional divergence methods. The sequence and structural alignments of these three PDB structures are shown in Appendix A. The computational methods used in this study integrate molecular evolution techniques and structural bioinformatics to assess functional divergence [20]. Functional divergence analysis was employed to identify amino acid residues likely subjected to selective pressures post-gene duplication, potentially influencing unique functional roles among ADH1 family members [11,21]. Network analysis, focusing on centrality metrics such as betweenness, closeness, and degree, enabled the examination of residue interactions within the three-dimensional ADH1 structures. These centrality metrics reveal residues critical to structural integrity and dynamic communication pathways within the enzyme, further elucidating their roles in substrate binding or catalysis. The integration of network centrality with functional divergence scores allows for a refined understanding of the specific evolutionary constraints that shape each ADH1 enzyme’s structural and functional landscape [22,23]. Computational methods help in understanding the molecular architecture and evolutionary forces driving functional differentiation in ADH1 homologs [24,25]

To investigate the structural stability and evolutionary patterns of alcohol dehydrogenase (ADH) enzymes, we employed several analytical methods. We conducted a phylogenetic analysis to discern the evolutionary relationships among ADH subtypes, emphasizing the conservation and divergence of key enzymes, particularly ADH1 and ADH3, across various taxa. Molecular dynamics (MD) simulations were then performed to explore the conformational dynamics and stability of ADH proteins over time. By applying Newtonian physics to model atomistic movements, these simulations provided detailed insights into structural fluctuations at high temporal resolution. Key measurements, such as root mean square deviation (RMSD), quantified the extent of structural deviations from the initial configuration, indicating protein stability. This approach enabled us to observe dynamic behaviors critical to understanding protein function and flexibility in a biological context. Additionally, we used dynamic cross-correlation maps (DCCM) to examine inter-residue dynamics, identifying correlated and anti-correlated motions that suggest functionally relevant domains. Lastly, network centrality analysis, focusing on betweenness, degree, and closeness measures, helped pinpoint residues critical for structural integrity and functional connectivity, allowing us to map essential nodes within the protein interaction network.

By leveraging these computational methods, our study aims to dissect the molecular architecture and evolutionary forces that have driven functional differentiation across ADH1 homologs, offering insights into both conserved and lineage-specific adaptations.

## 2. Materials and Methods

### 2.1. Phylogenetic Analysis

The phylogenetic tree was generated using the Diverge version 3.0 program [26,27], with the NJ (Neighbor-Joining) algorithm [28] employing the p-Distance method for distance estimation. Visualization was performed using the iTOL (Interactive Tree Of Life) program [29]. The NJ algorithm constructs a distance matrix by measuring the evolutionary distances between sequences, then iteratively combines the closest sequences into new clusters. The distances between these new clusters and the remaining sequences are recalculated, and this process is repeated until all sequences are represented in the form of a phylogenetic tree. A total of 139 ADH gene sequences were utilized to construct phylogenetic trees for each species and ADH type. 

### 2.2. Molecular Dynamics Simulation

We performed MD simulations using the NAMD package—version 2.14 [30] with the CHARMM36 force field [31] and with protein parameters incorporating the CMAP corrections [32] for three PDB structures (1CDO, 1HDX, and 4W6Z). The TIP3P water model [33] was employed. The particle mesh Ewald (PME) method was used with a direct space cut-off of 12 Å [34]. The damping coefficient for the Langevin dynamics simulation was 5 ps−1. The Nosé–Hoover method was used to maintain constant pressure (1 atm) [35]. MD simulation was performed in the NPT ensemble at 310 K. We performed MD simulation for 100ns for each of the three PDB structures.

### 2.3. Network Analysis

The dynamic cross-correlation map (DCCM) method was used to analyze the correlation between residues in trajectories of MD simulations [36,37,38].
(1)$$C_{ij}=\frac{<\left(r_{i}\left(t\right)-<r_{i}\left(t\right)>\right)\left(r_{j}\left(t\right)-<r_{j}\left(t\right)>\right)>}{\sqrt{\left(<r_{i}^{2}\left(t\right)>-<r_{i}\left(t\right){>}^{2})(<r_{j}^{2}\left(t\right)>-<r_{j}\left(t\right){>}^{2}\right)}}$$
where *r_i_*(*t*) and *r_j_*(*t*) are the atomic positions of the *i*-th and *j*-th Cα atoms at time t. The quantity *r_i_*(*t*) − <*r_i_*(*t*)> corresponds to the fluctuation of the *i*-th atom and *r_j_*(*t*) − <*r_j_*(*t*)> to that of the *j*-th atom. A correlation map for the conformational changes for all the Cα atoms during the 100 ns MD simulation was obtained. The *C_ij_* value (Equation (1)) in the DCCM is an adjacency matrix. In the constructed network, each node corresponds to a Cα atom, and each edge is an information transfer probability (i.e., cross-correlation). The weight *w_ij_* of the correlation edge between the nodes *i* and *j* was defined as [39,40]
(2)$$\;w_{ij}=-\log\left|C_{ij}\right|$$

To identify and quantify the nodes that occupy critical positions in a network, several centrality measures were proposed—including the betweenness, closeness, and degree centralities [41,42,43]—and were used in our analysis. The closeness centrality is defined as the average length of the shortest paths between a node and all the other nodes in a network. This is a measure of how long it will take information to spread from one given node to the other nodes. The closeness centrality is calculated as
(3)$$C_{C}\left(v_{i}\right)=\frac{n-1}{\sum_{j\neq i}g\left(v_{i},v_{j}\right)}$$
where *g* (*v_i_*, *v_j_*) is the shortest path with a weight between nodes *i* and *j*. *n* is the number of nodes in the graph. The betweenness centrality is a measure of how many information pathways flow through a node in a network. The betweenness of node *i* is the fraction of the shortest paths between pairs of nodes that pass through this node. The betweenness centrality is calculated as
(4)$$b_{i}=\frac{\sum_{s<t}g_{i}^{st}/n_{st}}{\frac{1}{2}n\left(n-1\right)}$$
where $g_{i}^{st}$ is the number of shortest paths from *s* to *t* with a weight that passes through node *i*, and *n_st_* is the total number of shortest paths from *s* to *t*.

The degree centrality measures the number of edges incident on a node in a network and is calculated as
(5)$$C_{D}\left(v_{i}\right)=\underset{j}{\sum }A_{ij}$$
where *A_ij_* is the adjacency matrix: if *w_ij_* > 0 then *A_ij_* = 1, otherwise *A_ij_* = 0.

### 2.4. Posterior Probability

Gene duplication is an important event of evolution for diversification. The divergence of ancestral genes by gene duplication leads to the generation of several multigene families (Figure 2a). Type-I functional divergence refers to the process in which duplicated genes evolve to have different functional divergences, often resulting in a change in the rate of evolution between the duplicates. This divergence is a key mechanism by which organisms gain new capabilities and adapt to their environment over time. A statistical method for the estimation of type-I functional divergence after gene duplication is well-defined [44]. After generating multiple alignments of amino acid sequences, the altered functional constraint between two gene clusters can be given by
(6)$$r_{\lambda }=\frac{Cov\left(\lambda_{1},\lambda_{2}\right)}{\sqrt{Var\left(\lambda_{1}\right)Var\left(\lambda_{2}\right)}}$$
where $Var\left(\lambda_{1}\right)$, $Var\left(\lambda_{2}\right)$, and $Cov\left(\lambda_{1},\lambda_{2}\right)$ are the variances and covariance of $\lambda_{1}$ and $\lambda_{2}$ [45]. λ1 and λ2 are evolutionary rates for class 1 and class 2. For no functional divergence after gene duplication, $r_{\lambda }=1$; otherwise, $r_{\lambda }<1$. The measure of type-I functional divergence is defined by
(7)$$\theta_{\lambda }=1-r_{\lambda }v$$

As functional divergence increases, $\theta_{\lambda }$ increases from 0 (functional constraint) to 1 (functional divergence). For the two-state model, all sites are categorized into two groups: functional constraint (*F*_0_) and functional divergence (*F*_1_) (Figure 2b,c). The prior probability for the site in the amino acid sequence is given by
(8)$$P\left(F_{1}\right)=1-r_{\lambda }=\theta_{\lambda },\;\;\;P\left(F_{0}\right)=1-\theta_{\lambda }$$

Amino acids showing significant rate differences between two gene clusters are very important for understanding the functional and structural evidence of molecular evolution. The posterior probability of each site is a critical measure for selecting amino acids showing significant rate differences after the gene duplication event. Gu [45] estimated the coefficient of functional divergence based on the Poisson model. When the number of amino acid changes is *X_i_* (*i* = 1,2 for gene cluster) at a given site, the probability of *X_i_* = *k* is given by
(9)$$p_{i}\left(k\right)=\frac{{\left(\lambda_{i}T_{i}\right)}^{k}}{k!}e^{-\lambda_{i}T_{i\;}}$$
where *T*_1_ and *T*_2_ are the total evolutionary times of clusters 1 and 2, respectively [45].

By the assumption that the evolutionary rate changes among sites according to gamma distribution, the distribution of evolution rate, *ϕ*(*λ*), is defined by
(10)$$\phi \left(\lambda \right)=\frac{\beta^{\alpha }}{\Gamma \left(\alpha \right)}\lambda^{\alpha -1}e^{-\beta \lambda }\;$$
where $\lambda =\lambda_{1}\;\mathrm{or}\;\lambda_{2}.$

Then, the joint distribution of the number of changes, *P*(*X*_1_ = *i*, *X*_2_ = *j*), is given by
(11)$$P(X_{1}=i,\;X_{2}=j\;|F_{1})=Q_{1\;}\left(i\right)Q_{2\;}\left(j\right),\;P(X_{1}=i,\;X_{2}=j\;|F_{0})=K_{12\;}\left(i,j\right)$$
where 1 and 2 are gene clusters. *Q*_1_ and *Q*_2_ are given as
(12)$$Q_{1}\left(i\right)=\int_{0}^{\infty }p_{1}\left(i\right)\phi \left(\lambda_{1}\right)d\lambda_{1},\;Q_{2}\left(j\right)=\int_{0}^{\infty }p_{2}\left(j\right)\phi \left(\lambda_{2}\right)d\lambda_{2}$$
The posterior probability of state *F*_1_ at the specific site with *X*_1_ and *X*_2_ changes in clusters 1 and 2 is given by [44]
(13)$$P\left(F_{1}|X_{1},\;X_{2}\right)=\frac{\theta_{\lambda }Q_{1\;}Q_{2}}{\left(1-\theta_{\lambda }\right)K_{12}+\theta_{\lambda }Q_{1\;}Q_{2}}$$

*Q* values are calculated using Diverge [27] for the type-I functional divergence in this study.

## 3. Results

### 3.1. Phylogenetic Analysis

The generated phylogenetic tree provides a detailed overview of the evolutionary relationships among various alcohol dehydrogenase (ADH) enzymes across a wide range of species. In the fish lineage, sequences diverged into ADH1 and ADH3, while the mammalian and fungal lineages branched into ADH1, ADH2, ADH3, ADH4, and ADH5. The mammalian ADH lineage reveals a more complex structure, with ADH1 further subdivided into several subclusters: ADH1A, ADH1B (PDB ID:1HDX), ADH1C, along with ADH2, ADH3, ADH4, and ADH5 forming a single cluster. Notably, ADH4 is positioned adjacent to ADH1, suggesting its origin from a gene duplication event of ADH1. In fungi, ADH1 (PDB ID: 4W6Z), ADH2, and ADH5 appear to cluster within the *Saccharomyces cerevisiae*, indicating a common ancestor that likely resulted from gene duplication of ADH1. Importantly, ADH2 is more closely related to ADH1 than to ADH5, implying a greater functional or structural similarity.

The ADH1 and ADH2 sequences from each species of *Lachancea* and *Kluyveromyces* were found to cluster together, whereas *Kluyveromyces* ADH3 and *Lachancea* ADH3 formed a separate cluster. This indicates that the genetic phylogenetic relationships among fungal ADH sequences differ from those of mammals, suggesting that each ADH type does not exist as a single cluster. This finding implies that although different species may share the same ADH type during the evolutionary process, they have undergone distinct genetic changes, resulting in functional differentiation and diverse characteristics.

Notably, ADH3 appears to be highly conserved, indicating its ancient origin and critical role across diverse taxa. It is present in a broad spectrum of species, from vertebrates like mammals, amphibians, and birds to invertebrates such as mollusks and nematodes. This conservation suggests that ADH3 plays a fundamental and possibly essential role in the metabolic processes of these organisms. 

In contrast, ADH1 and ADH2 show significant evolutionary divergence, particularly within mammalian lineages, where multiple subtypes have emerged. This divergence suggests that these enzymes have undergone specialization to fulfill specific metabolic functions, such as alcohol metabolism, retinoid metabolism, or detoxification pathways, which are more complex in higher vertebrates. The presence of *Viridiplantae* ADH and fungi ADH as outgroups helps root the tree and provides a broader evolutionary context, allowing for a more comprehensive understanding of the evolutionary history and diversification of ADH enzymes (Figure 3). The branch lengths in the tree, which represent evolutionary distances, indicate varying levels of divergence, with longer branches reflecting greater evolutionary separation. This variation emphasizes the genetic diversity among the ADH enzymes and highlights the different evolutionary pressures that have shaped their development across species.

ADH1 is particularly significant due to its greater evolutionary divergence and specialization compared to other ADH types. While ADH3 is highly conserved across a wide range of species, ADH1 has evolved distinctively, particularly within mammals, amphibians, and birds. This evolutionary divergence suggests that ADH1 has adapted to meet specific metabolic requirements, such as alcohol metabolism, detoxification, and other specialized biochemical processes that are crucial for the survival and adaptation of these organisms. The evolution of ADH1 into different subtypes within mammals further underscores its importance, indicating a role in species-specific metabolic pathways. This makes ADH1 a critical enzyme for studying the evolutionary adaptations and functional specialization of metabolic processes across different species.

In summary, these results indicate that a single ADH type can diversify into subgroups with varying genetic mutations or structural differences rather than remaining as a singular cluster within the same classification.

### 3.2. RMSD Analysis

Root mean square deviation (RMSD) values are used to evaluate the structural stability and conformational changes of proteins during molecular dynamics (MD) simulations. Figure 4 shows the RMSD data for three pdb structures 1CDO, 1HDX, and 4W6Z. The thick line indicates the averages of the RMSD values. These RMSD values were calculated for a period of 100ns MD simulation. 

For 1CDO, the RMSD starts at a low value, showing that the protein initially stays close to its reference structure. During the first 20 ns, the RMSD increases quickly and then stabilizes around 3.0 Å. This indicates that 1CDO undergoes early conformational changes but eventually reaches a stable structure, with RMSD values consistently within the 5 Å range. Similarly, 1HDX shows an initial rise in RMSD, with values increasing until about 20–30 ns, where it stabilizes around 3.0 Å. However, 1HDX exhibits slightly more variability than 1CDO, suggesting that while the protein also reaches a stable conformation, it still experiences minor fluctuations during the simulation. The larger fluctuations in 1HDX’s RMSD compared to 1CDO indicate more significant conformational changes before stabilization, though overall stability is comparable.

In contrast, 4W6Z shows greater fluctuations throughout the simulation, particularly between 50 and 100 ns, with RMSD values varying between 2.5 and 3.5 Å. This suggests that 4W6Z is less stable or more flexible than the other two proteins. The continuous fluctuations imply that 4W6Z does not settle into a single stable conformation but instead transitions between multiple conformations or remains dynamic throughout the simulation. The RMSD profile for 4W6Z also shows distinct phases, with an initial increase followed by periods of stability and then another increase towards the end, indicating multiple stages of conformational changes. The root mean square fluctuation (RMSF) values during 100 ns MD simulations repeated three times for three PDB structures shown in Appendix A. We found that two loop regions for all three proteins show relatively higher fluctuation: for 1CDO (PHE 94–PRO 121 and LEU 298–GLY 311), 1HDX (GLY 98–ARG 128 and SER 298–ARG 312), and 4W6Z (SER 96–PRO112 and MET 270–ILE 288).

In summary, the MD simulations reveal that all three proteins undergo conformational changes, but 1CDO and 1HDX reach stable conformations after an initial period of adjustment. In contrast, 4W6Z remains more dynamic, suggesting greater flexibility or lower stability. These differences may be related to the proteins’ functional roles, interactions with their environment, or inherent structural properties. The dynamic behavior of 4W6Z might imply a need for flexibility in its function, while the stability of 1CDO and 1HDX suggests more rigid biological roles.

### 3.3. Dynamic Cross-Correlation Analysis

Dynamic cross-correlation analysis was performed for the holo-proteins of the PDB structures 1CDO (*Gadus morhua*), 1HDX (*Homo sapiens*), and 4W6Z (*Saccharomyces cerevisiae*) (Figure 5a–c). 

For 1CDO, the DCCM showed a complex pattern of correlations, with very sparse red patterns. This suggests that the movements of the residues within this protein are intricately linked, with only some residues moving in a highly correlated manner (as indicated by the red points), while others show anti-correlation in their movements (blue points). The DCCM for 1HDX showed prominent red areas, suggesting a strong correlation between residues. This could indicate a dynamic behavior different from that of 1CDO, with certain parts of the protein moving in a more coordinated manner, especially among residues 1–100 and near the central diagonal line. The DCCM of 4W6Z had patterns of residue correlations distinct from those in the other maps. This suggests that despite the structural similarities with the other two proteins, 4W6Z has unique dynamic properties. The red areas were more prominent in the 4W6Z structure than in the other two structures. The distinct shape of the protein might also influence these dynamic correlations. The strong correlations could be indicative of functional domains within the protein that work together and move in a coordinated fashion to achieve the protein’s function.

### 3.4. Principal Component Analysis

Principal component analysis (PCA) is a statistical procedure that uses an orthogonal transformation to convert a set of observations of possibly correlated variables into a set of values of linearly uncorrelated variables called principal components [46]. The PCA plots of 1CDO, 1HDX, and 4W6Z are shown in Figure 6. For 1CDO, the first principal component (PC1) explained 28.15% of the variance, while the second principal component (PC2) explained 12.22%. For 1HDX, PC1 explained 29.56% of the variance, while PC2 explained 13.75%. For 4W6Z, PC1 explained a significant 49.68% of the variance, while PC2 explained 11.65%. The PCA results suggest a complex interplay of movements within the protein structure, with certain residues showing high correlation, implying coordinated movement or interaction.

When we look at how the path of movement is shown on a graph, we see a shape like half a circle or the letter U for all three proteins (Figure 6). Such a pattern could imply that the proteins are sampling a range of conformations. The half-circle or U shape might represent a conformational space where the proteins are transitioning between different conformational states. This pattern also likely suggests random diffusion within the simulation, which can be interpreted as thermal motion across a flat, gently sloping landscape of free energy [47,48,49]. Figure 6 also shows the resulting PC analysis scree plot, indicating the proportion of variance against its eigenvalue rank. This implies how much of the data’s overall structure and variation can be captured by considering just the first few principal components. For 1CDO, the first three PCs together explained 47% of the variance. For 1HDX, they explained 50% of the variance. For 4W6Z, they explained 68% of the variance. For 4W6Z, a significant majority (68%) of the data’s behavior can be understood by analyzing the first three PCs, indicating that these components are very informative. In contrast, for 1CDO and 1HDX, the first three PCs are less informative, capturing less than half of the total variance [50].

### 3.5. Centrality Measures

In the context of network analysis, the concept of centrality is instrumental in discerning the importance of a node (or residue) and an edge’s (or interaction’s) role in network connectivity and the propagation of information within it. Two primary metrics in this study were closeness centrality and betweenness centrality. Closeness centrality serves as an estimate of the speed at which information can disseminate through a node to other nodes, taking into account the shortest paths. It essentially measures the average length of the shortest paths from a node to all other nodes in the network. Betweenness centrality is used to pinpoint nodes that lie on the most direct paths of communication, thereby controlling the flow of information. It quantifies the frequency at which a node appears on all shortest paths between two other nodes, thus highlighting its role as a “bridge” in the network. By integrating centrality measures, we can deepen our understanding of the dynamics of the network. This integrated approach aids in identifying critical elements that are integral to maintaining the network’s optimal functioning and integrity. 

The top 15 residues with the highest betweenness, closeness, and degree centralities are listed in Table 1, Table 2 and Table 3. ALA 207, PHE 230, LEU 206, and ASN 278 in 1CDO (Table 1); LEU 279, SER 289, GLU 267, and MET 276 in 1HDX (Table 2); and LEU 116, ASN 94, HIS 240, LEU 93, LYS 160, and TRP 92 in 4W6Z (Table 3) are the residues that have both high betweenness and closeness. The high values suggest that these residues play an important role in the protein’s structural network.

Residues that have high betweenness, closeness, and degree centrality measures in a protein structure are often considered important due to their potential role in the protein’s function, stability, and interactions: 207 ALA for 1CDO, 279 LEU, 267 GLU/276 MET for 1HDX, and 116 LEU /160 LYS for 4W6Z, positioned close to the ligand. They often reside in key positions that connect different parts of the protein, acting as “hubs” or “bridges” in the protein network. High-centrality residues are often found in or near the active site of enzymes or the binding site of receptors (Figure 7). They may be directly involved in the protein’s function or mediate catalytic reactions with other molecules. Mutations at these positions are more likely to disrupt the protein’s structure and function than other mutations [51].

### 3.6. Type-I Functional Divergence

To assess the Type-I functional divergence in the three proteins, we analyzed the relationship between Q-values and three key centrality measures: betweenness, closeness, and degree. The maps presented (left to middle panels) display the distribution of normalized Q-values against these centrality measures for three structures (Figure 8 (a) 1CDO, (b) 1HDX, and (c) 4W6Z). The results show a stronger correlation between closeness centrality and Q-values, as seen in the dense clustering of high-Q-value regions in the closeness maps. This suggests that residues with higher closeness centrality, which are more efficiently connected to other residues, play a key role in functional divergence. In contrast, betweenness centrality shows a sparse distribution of Q-values, indicating that residues on key paths between other residues may not contribute as strongly to divergence. The degree centrality maps also show some correlation with Q-values, but not as significantly as closeness centrality.

Several key residues that exhibit both high closeness centrality and Q-values were identified and highlighted in the 3D structural representations. For the fish ADH1/ADH3 divergence (ADH1: 1CDO), these include ILE 36, PHE 54, SER 178, ALA 203, VAL 204, ALA 208, VAL 209, VAL 223, GLU 231, LYS 232, VAL 235, ASP 240, PHE 241, ASN 243, ILE 251, SER 252, VAL 275, ASN 356, and ILE 359 (Appendix A). In the case of fungi ADH1/ADH5 divergence (ADH1: 4W6Z), the key residues are ILE 156, LEU 167, MET 168, HIS 171, and VAL 242 (Appendix A). For the Mammalia ADH1/ADH4 divergence (ADH1: 1HDX), important residues include ASP 49, HIS 51, PRO 62, PHE 130, HIS 138, PHE 146, GLY 202, GLU 252, VAL 268, and ARG 271 (Appendix A). These residues are central to the protein structure and likely crucial for maintaining protein function, making them hotspots for functional divergence. This analysis suggests that closeness centrality is strongly correlated with type-I functional divergence, helping to identify evolutionary important sites within protein structures.

## 4. Discussion

In this study, we investigated the type-I functional divergence of the ADH1 family in three species—*Saccharomyces cerevisiae* (PDB id: 4W6Z), *Gadus morhua* (PDB id: 1CDO), and *Homo sapiens* (PDB id: 1HDX)—using a combination of phylogenetic analysis, molecular dynamics (MD) simulations, and network analysis. Through posterior probability estimation, we identified residues showing significant functional divergence, underscoring the evolutionary adaptations that differentiate these proteins across species.

Our network analysis, based on centrality measures derived from MD simulations, revealed distinct evolutionary patterns in the ADH1 proteins. Specifically, in 1CDO, residues Leu298 and Lys232 exhibited both high betweenness and closeness centrality, correlating strongly with high posterior probabilities for functional divergence. Similarly, His51 in 1HDX demonstrated a significant correlation between high centrality measures and functional divergence, emphasizing its role in catalysis, likely due to its proximity to NAD+ binding sites. These findings suggest that key catalytic and binding residues, which act as structural hubs, may undergo functional diversification during evolution, facilitating novel functional capabilities.

Interestingly, the absence of residues with both high centrality and functional divergence in 4W6Z points to a more conserved evolutionary pathway in *S. cerevisiae*. This indicates that while structural and functional divergence occurs in the ADH1 family, it may be species-specific, shaped by differing evolutionary pressures. This conserved nature in 4W6Z could imply that functional divergence in *S. cerevisiae* is limited to non-catalytic regions or that it is driven by factors not captured by centrality measures in our analysis.

The correlation observed between functional divergence and network centrality measures such as closeness highlights the evolutionary importance of structural “hub” residues. Residues central to the network tend to play pivotal roles in maintaining protein stability and function. Presumably, evolutionary changes at these sites may drive significant functional shifts. These insights provide a framework for further exploring the relationship between protein structure, functional divergence, and evolution across species.

## 5. Conclusions

This study offers an analysis of type-I functional divergence in the ADH1 families of *Saccharomyces cerevisiae*, *Gadus morhua*, and *Homo sapiens*. By integrating phylogenetic analysis, MD simulations, and network theory, we identified critical residues that have undergone functional divergence. The strong association between centrality measures (closeness) and functionally divergent residues, particularly in *Gadus morhua* and *Homo sapiens*, underscores the importance of structural hubs in protein evolution.

Our results suggest that evolutionary divergence at these “hub” residues has likely contributed to the diversification of ADH1 functions, especially in catalysis and NAD+ binding. The conserved nature of the *S. cerevisiae* ADH1 structure, as evidenced by the lack of functionally divergent residues despite high centrality, suggests that different evolutionary pressures may guide functional divergence in this species.

These findings not only enhance our understanding of the molecular evolution of the ADH1 family but also provide valuable insights for future studies investigating evolutionary changes in other enzyme families. The identification of key structural residues that drive functional shifts offers potential applications in enzyme engineering, drug design, and the broader study of protein evolution.

## Figures and Tables

**Figure 1 biomolecules-14-01473-f001:**
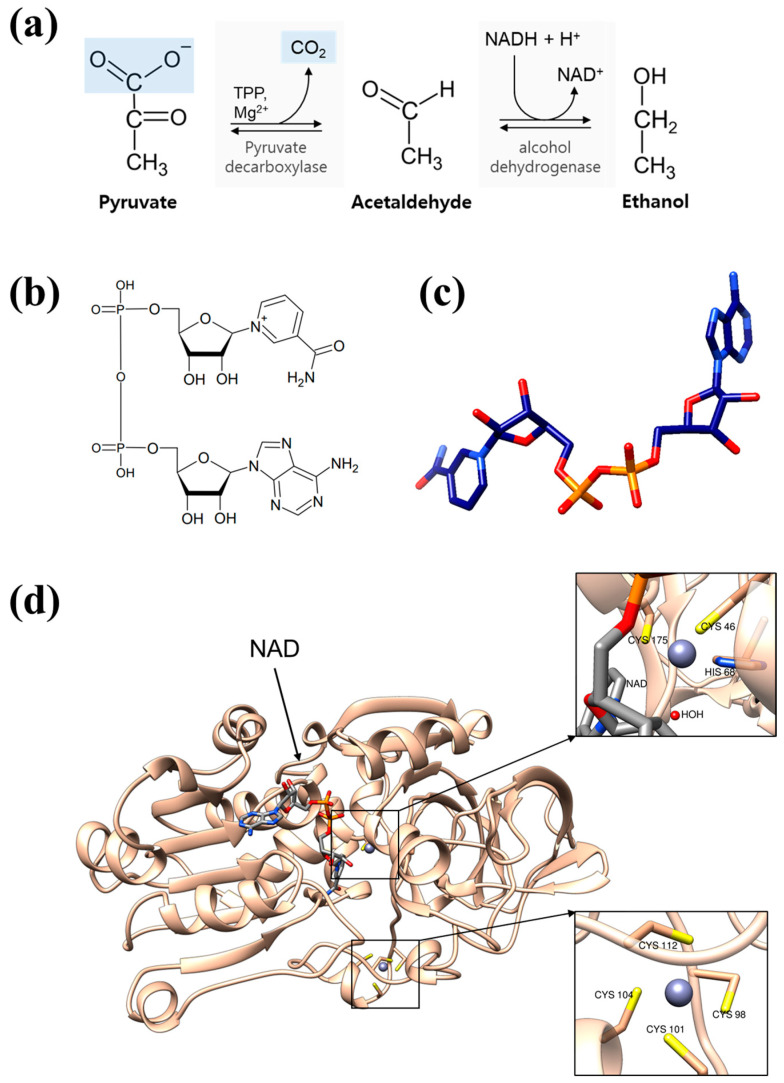
Alcohol dehydrogenase is an enzyme used together with pyruvate decarboxylase in alcoholic fermentation to convert NADH to NAD+ while converting acetaldehyde to ethanol. (**a**) The conversion process. (**b**) 2D and (**c**) 3D structures of NAD. (**d**) Structure of human ADH in complex with NAD (PDB id: 1HDX). The zinc ions are shown in the boxes.

**Figure 2 biomolecules-14-01473-f002:**
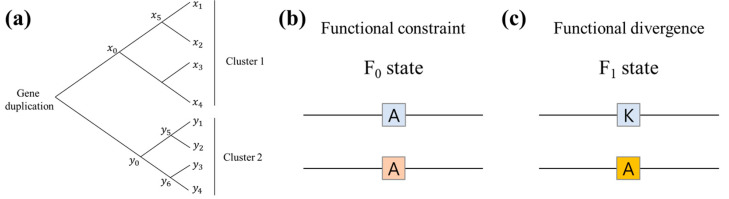
(**a**) Emergence of gene families after gene duplication and formation of clusters. (**b**) A scheme of the F0 state (functional constraint). (**c**) A scheme of the F1 state (functional divergence).

**Figure 3 biomolecules-14-01473-f003:**
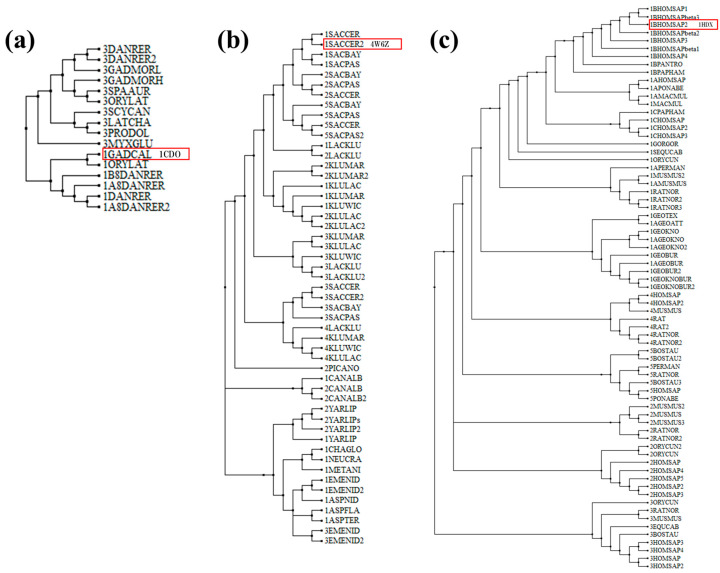
Phylogenetic tree of alcohol dehydrogenase (ADH) enzymes of (**a**) ADH1/ADH3, (**b**) ADH1/ADH2/ADH3/ADH4/ADH5, (**c**) ADH1/ADH2/ADH3/ADH4/ADH5. The entities in the phylogenic tree crystalized are shown in the red box.

**Figure 4 biomolecules-14-01473-f004:**
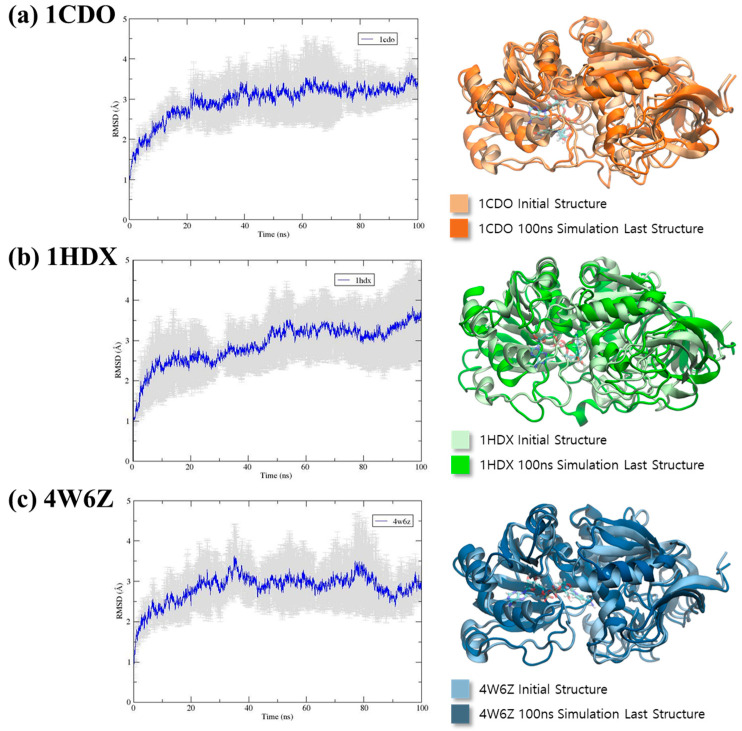
RMSD values and standard deviations of 100 ns MD simulations, repeated three times for three PDB structures and the alignment of the initial structures and final snapshot (100 ns) structures, respectively. (**a**) RMSD values and standard deviations of 1CDO. (**b**) RMSD values and standard deviations of 1HDX. (**c**) RMSD values and standard deviations of 4W6Z.

**Figure 5 biomolecules-14-01473-f005:**
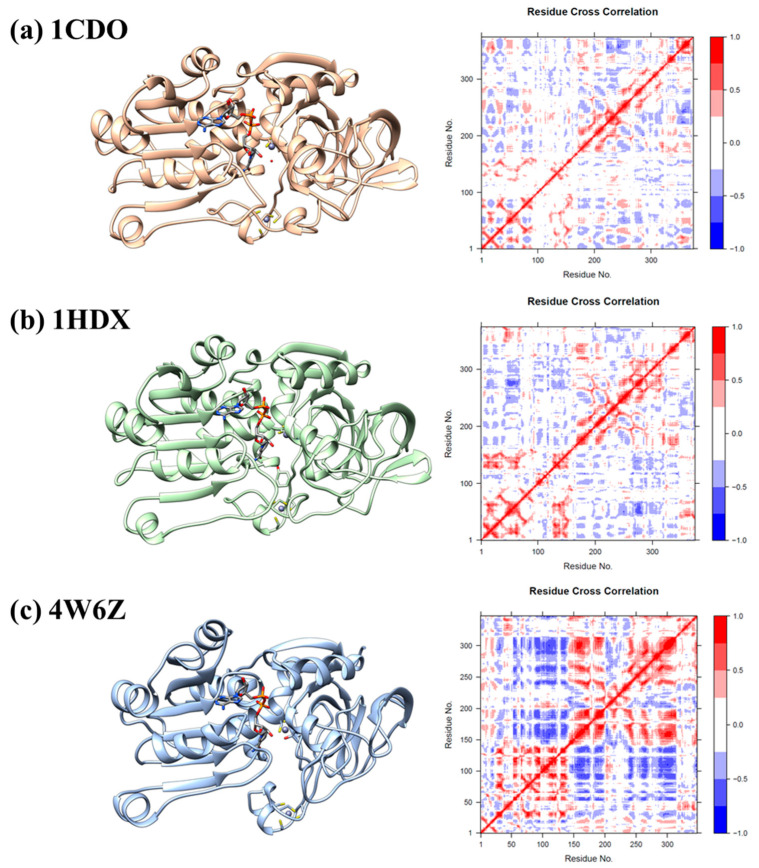
Dynamic cross-correlation map (DCCM) calculations throughout 100 ns molecular dynamics simulation of a protein. The maps provide information about the dynamic correlation between the Cα atoms of amino acids. Correlation between residue fluctuations: blue, negative; red, positive correlation. The color coding from blue to red indicates the degree of correlation, with red representing areas of high correlation which often implies coordinated movement or interaction between those residues. The diagonal line of the DCCM represents the correlation of each residue with itself, which is always perfect (hence the line is red). The points not on the diagonal line represent the correlations between different residues. (**a**) Dynamic cross-correlation map of 1CDO. (**b**) Dynamic cross-correlation map of 1HDX. (**c**) Dynamic cross-correlation map of 4W6Z.

**Figure 6 biomolecules-14-01473-f006:**
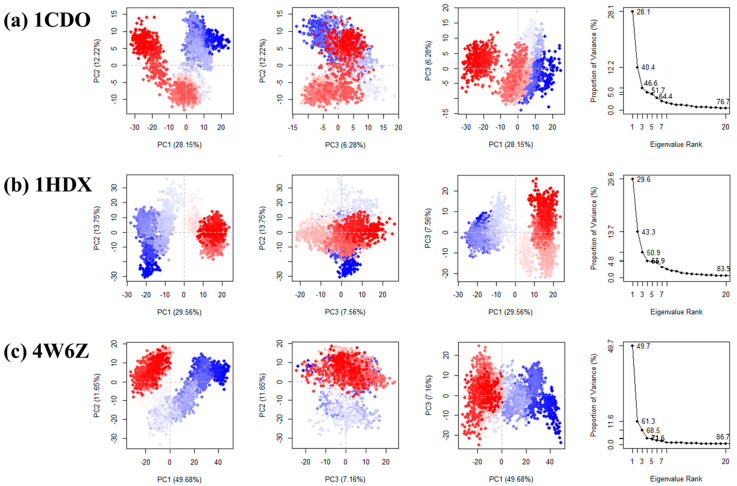
Principal component analysis throughout the 100 ns molecular dynamics simulation of each protein. The plots of PC2 vs. PC1 show the correlation between the first and second principal components, etc. Each point in the scatter plot represents a different conformation of the protein during the simulation. If two points are close together in the plot, it means that those two conformations are similar in terms of the movements of the residues. The percentage values given in the scatter plots represent the variance explained by each principal component. (**a**) Principal component analysis of 1CDO. (**b**) Principal component analysis of 1HDX. (**c**) Principal component analysis of 4W6Z.

**Figure 7 biomolecules-14-01473-f007:**
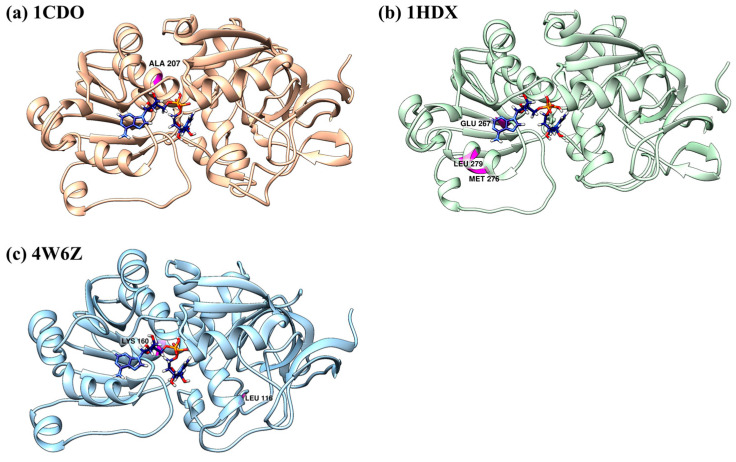
Positions of residues with high betweenness centrality, closeness centrality, and degree centrality in the 3D structure of (**a**) 1CDO, (**b**) 1HDX, and (**c**) 4W6Z.

**Figure 8 biomolecules-14-01473-f008:**
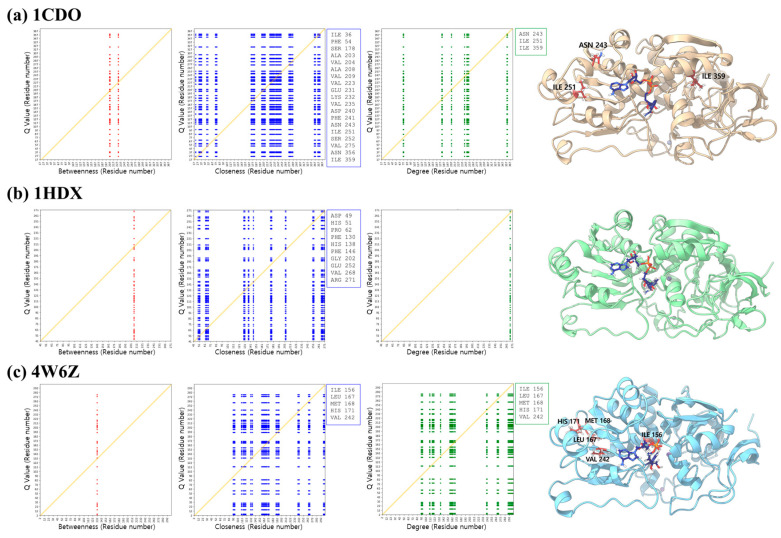
Correlation between Q values and centrality measures betweenness, closeness, and degree in (**a**) 1CDO, (**b**) 1HDX, and (**c**) 4W6Z. The amino acid residues on the diagonal lines (orange color) are shown in the boxes.

**Table 1 biomolecules-14-01473-t001:** Top 15 residues with high betweenness, closeness, and degree in 1CDO. The residues that overlap in both betweenness, closeness, and degree are shown in bold.

	Betweenness	Closeness	Degree
1	**ALA 207**	**ALA 207**	SER 76
2	PHE 230	PHE 181	PRO 250
3	LEU 206	LYS 366	LEU 361
4	CYS 104	LEU 361	ILE 251
5	ASP 337	LEU 206	VAL 77
6	LYS 285	PHE 230	PHE 181
7	THR 347	MET 362	ASN 243
8	GLU 27	MET 231	GLY 182
9	ASN 278	ASN 243	GLU 249
10	GLU 35	ASN 278	**ALA 207**
11	LEU 172	GLY 182	ILE 359
12	ARG 130	ALA 358	ALA 358
13	LEU 298	LYS 232	VAL 254
14	ILE 156	GLY 365	LYS 366
15	GLU 281	ASP 357	VAL 30

**Table 2 biomolecules-14-01473-t002:** Top 15 residues with high betweenness, closeness, and degree in 1HDX. The residues that overlap in both betweenness, closeness, and degree are shown in bold.

	Betweenness	Closeness	Degree
1	**LEU 279**	MET 275	**GLU 267**
2	SER 206	**MET 276**	THR 274
3	PHE 335	THR 274	**MET 276**
4	SER 289	**LEU 279**	ARG 129
5	ARG 37	SER 278	MET 275
6	ASN 118	ALA 277	ALA 12
7	**GLU 267**	**GLU 267**	GLY 66
8	GLY 66	LEU 280	GLY 270
9	THR 131	ASP 273	SER 278
10	ARG 129	LEU 272	**LEU 279**
11	HIS 51	ARG 271	ASP 49
12	GLY 117	PHE 266	PHE 130
13	ILE 137	SER 289	PHE 146
14	ARG 47	VAL 13	PHE 266
15	**MET 276**	CYS 282	THR 131

**Table 3 biomolecules-14-01473-t003:** Top 15 residues with high betweenness, closeness, and degree in 4W6Z. The residues that overlap in both betweenness, closeness, and degree are shown in bold.

	Betweenness	Closeness	Degree
1	ASP 132	LEU 93	ASN 297
2	**LEU 116**	**LEU 116**	THR 301
3	ASN 94	TRP 92	**LYS 160**
4	HIS 240	ASN 94	VAL 266
5	LEU 93	LYS 163	GLY 296
6	ARG 196	LEU 162	ASP 300
7	ILE 288	HIS 240	ARG 302
8	CYS 111	**LYS 160**	CYS 111
9	TYR 189	SER 117	ILE 156
10	**LYS 160**	LEU 167	TYR 294
11	VAL 266	TYR 159	ARG 298
12	GLU 67	ASP 115	ALA 161
13	ASN 110	ALA 161	ALA 299
14	TYR 195	HIS 171	TYR 159
15	TRP 92	VAL 173	**LEU 116**

## Data Availability

The raw data supporting the conclusions of this article will be made available by the authors on request.

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
