# Peer review of "Computational Study of Network and Type-I Functional Divergence in Alcohol Dehydrogenase Enzymes Across Species Using Molecular Dynamics Simulation"

_biomolecules, 2024, doi:10.3390/biom14111473_

Round 1
Reviewer 1 Report
Comments and Suggestions for Authors
The manuscript titled “Computational Study of Network and Type-I Functional Divergence in Alcohol Dehydrogenase Enzymes Across Species using Molecular Dynamics Simulation” aimed to investigate the molecular evolution of three ADH1 families (Yeast alcohol dehydrogenase I, Cod(fish) alcohol dehydrogenase I, and human alcohol dehydrogenase I)) using network analysis and functional divergence methods. The manuscript is more like a preliminary research work and the data of the manuscript could not support the conclusion well, so that the manuscript is not suitable for publication currently.
The authors need to further summarize the necessity of studying the molecular evolution of ADH1 in the part of introduction.
For the phylogenetic analysis, the author only stated the relevant results and needs to further summarize the relevant conclusions and the reasons leading to this result.
It is necessary for the authors to compare the differences between the three ADH1s in detail in terms of structure and sequence, which contributes to study molecular evolution of ADH1.
The authors should clearly explain how the molecular dynamics results illustrate the evolution of the ADH1 molecule. The result of RMSD, DCCM and PCA were not enough, and the author should provided more simulation data.
The simulation time of 100 ns cannot guarantee rich sampling of the simulation system. The authors should extend the simulation time and perform multiple repeated simulations to avoid randomness in the simulation results.
Author Response
The manuscript titled “Computational Study of Network and Type-I Functional Divergence in Alcohol Dehydrogenase Enzymes Across Species using Molecular Dynamics Simulation” aimed to investigate the molecular evolution of three ADH1 families (Yeast alcohol dehydrogenase I, Cod(fish) alcohol dehydrogenase I, and human alcohol dehydrogenase I)) using network analysis and functional divergence methods. The manuscript is more like a preliminary research work and the data of the manuscript could not support the conclusion well, so that the manuscript is not suitable for publication currently.
We would like to appreciate the reviewer’s comments very much. We did our best to answer the questions and to revise our manuscript according to the comments. Thanks to the reviewer’s comments, our manuscript could be improved significantly.
Comment 1: The authors need to summarize further the necessity of studying the molecular evolution of ADH1 in the part of the introduction.
Answer: As suggested by the reviewer, we added the necessity of studying the molecular evolution of ADH1 and the introduction part and have revised the introduction section.
Comment 2: For the phylogenetic analysis, the author only stated the relevant results and needs to further summarize the relevant conclusions and the reasons leading to this result.
Answer: As suggested by the reviewer, we summarized the phylogenic analysis and result in the main text. (Line 162- 187) in main text.
Comment 3: It is necessary for the authors to compare the differences between the three ADH1s in detail in terms of structure and sequence, which contributes to study molecular evolution of ADH1.
Answer:As suggested by the reviewer, we compared the differences between the three ADH1s in detail in terms of structure and sequence in the main text (Line 107-108). The figure is shown in supplementary Figure S1.
Comment 4: The authors should clearly explain how the molecular dynamics results illustrate the evolution of the ADH1 molecule. The result of RMSD, DCCM and PCA were not enough, and the author should provided more simulation data.
Answer:It is not possible to show that MD simulation results illustrate the evolution of the ADH1 directly. Rather, in this study, we attempt to show that the functional divergence residues of the ADH1 are correlated with the key “hub” residues in the ADH1. To show that, we performed a structural network analysis of three ADH1 proteins employing MD simulation.
Comment 5: The simulation time of 100 ns cannot guarantee rich sampling of the simulation system. The authors should extend the simulation time and perform multiple repeated simulations to avoid randomness in the simulation results.
Answer: We agree with the reviewer that 100 ns simulation does not guarantee rich sampling of the simulation system. But 100 ns simulation provides information on the side chain fluctuations without significant structural change of backbones to some extent. 100 ns simulation is short to observe a conformational change of backbone at a large scale. But as shown in Figure 6 in the main text, the movements of the backbone of three structures (1CDO, 1HDX, 4W6Z) in the principal component axes (PC1 vs PC2, PC2 vs PC3, PC1 vs PC3) show different patterns, implying transitioning between different conformational states. A different movement of the backbone for three structures is also shown in DCCM patterns. It means that we can extract information on the interaction between backbone atoms of protein. Observing the unfolding process starting from the X-ray structure is not our purpose. We attempted to observe the backbone dynamics around global minimum in the presence of ligand (NAD+) and find the interactions between backbone atoms during 100 ns. As suggested by the reviewer, we extended one of the proteins (IHDX) up to 300 ns with repeated simulations (two times) and found that the RMSD values are very stable. Up to 3oons, we did not found any significant changes in the backbone RMSD.

Reviewer 2 Report
Comments and Suggestions for Authors
The manuscript discusses the use of molecular dynamics simulation and network theory to investigate the molecular evolution of alcohol dehydrogenase (ADH) enzymes from Saccharomyces cerevisiae, Gadus morhua, and Homo sapiens. The authors discovered a correlation between amino acid residues with high betweenness and closeness and type-I functional divergence of ADH.
The topic of the manuscript is interesting in the field of molecular biology, and the computational aspect appears to be well-founded. There are some minor issues that are not clearly stated, but they do not significantly affect the overall positive impression of this work.
Specifically:
1) The RMSD analysis depicted in Fig. 4 provides valuable information about the conformational stability of the studied enzymes. However, it would be beneficial if the authors included a comparison of the initial and final MD sampled conformation of these enzymes to illustrate the structure overlap.
2) To assess the similarity between the three enzymes, as shown in Fig. 5, the inclusion of RMSF plots would be highly informative.
Author Response
Reviewer 2:
The manuscript discusses the use of molecular dynamics simulation and network theory to investigate the molecular evolution of alcohol dehydrogenase (ADH) enzymes from Saccharomyces cerevisiae, Gadus morhua, and Homo sapiens. The authors discovered a correlation between amino acid residues with high betweenness and closeness and type-I functional divergence of ADH.
The topic of the manuscript is interesting in the field of molecular biology, and the computational aspect appears to be well-founded. There are some minor issues that are not clearly stated, but they do not significantly affect the overall positive impression of this work.
We would like to appreciate the reviewer’s comments very much. We did our best to answer the questions and to revise our manuscript according to the comments. Thanks to reviewer’s comments, our manuscript could be improved significantly. Thank you for your thorough review and positive feedback on our manuscript. Your recommendation to accept the manuscript for publication is truly appreciated.
Comment 1: The RMSD analysis depicted in Fig. 4 provides valuable information about the conformational stability of the studied enzymes. However, it would be beneficial if the authors included a comparison of the initial and final MD sampled conformation of these enzymes to illustrate the structure overlap.
Answer: As suggested by the reviewer, we compared the initial and the final MD simulation conformation and improved Figure 4 with the comparison and structural overlap. (Line 332 in the main text)
Comment 2: To assess the similarity between the three enzymes, as shown in Fig. 5, the inclusion of RMSF plots would be highly informative.
Answer: As suggested by the reviewer, we calculated RMSF values for three proteins. We added RMSF values in the main text (Line 357-363). The figure is shown in Supplementary Figure S2.
Reviewer 3 Report
Comments and Suggestions for Authors
I am very grateful you for the invitation to review the manuscript biomolecules-3263909 by park and coauthors “Computational Study of Network and Type-I Functional Divergence in Alcohol Dehydrogenase Enzymes Across Species using Molecular Dynamics Simulation”. The work is interesting but needs adjustments to increase the quality of the material.
Comments:
· The abstract must deal with the entire work. Loose sentences, without context or any concordance. The summary needs to be redone.
· The computational method also needs to be included in the introduction. Currently, several studies already address it as the main theme of their work, and there is enough literature to incorporate it into this section of the paper.
· Improve Figure 4, which is difficult to understand.
· Insert conclusion.
· The work needs to be better discussed with the literature, a few studies referenced, with few comparisons with previous studies already published.
· According to Ithentic, the work shows 23% similarity with literature; this rate needs to be reduced to 15% at most.
Author Response
I am very grateful you for the invitation to review the manuscript biomolecules-3263909 by park and coauthors “Computational Study of Network and Type-I Functional Divergence in Alcohol Dehydrogenase Enzymes Across Species using Molecular Dynamics Simulation”. The work is interesting but needs adjustments to increase the quality of the material.
We would like to appreciate reviewer’s comments very much. We did our best to answer the questions and to revise our manuscript according to the comments. Thanks to reviewer’s comments, our manuscript could be improved significantly. Thank you for your thorough review and positive feedback on our manuscript. Your recommendation to accept the manuscript for publication is truly appreciated.
Comment 1: The abstract must deal with the entire work. Loose sentences, without context or any concordance. The summary needs to be redone.
Answer: We would like to appreciate reviewer’s comments very much. As suggested by the reviewer, we have revised the abstract section of the manuscript.
Comment 2: The computational method also needs to be included in the introduction. Currently, several studies already address it as the main theme of their work, and there is enough literature to incorporate it into this section of the paper.
Answer: We would like to appreciate the reviewer’s comments very much. As suggested by the reviewer, we have included the computational methods in the introduction and revised the introduction section with similar studies. It is shown in the main text (Line 125-145)
Comment 3: Improve Figure 4, which is difficult to understand.
Answer: As suggested by the reviewer, we improved Figure 4. The thick blue line is the average RMSD with standard deviations (shown as gray). Also, in Figure 4, we aligned the initial and final structures (100 ns) of MD simulation. It is shown in the main text (Line 332).
Comment 4: Insert conclusion.
Answer: We would like to appreciate the reviewer’s comments very much. As suggested by the reviewer, we have added a conclusion section.
Comment 5: The work needs to be better discussed with the literature, a few studies referenced, with few comparisons with previous studies already published.
Answer: We would like to appreciate reviewer’s comments very much. As suggested by the reviewer, we have added previous studies and compared them to the current study in the introduction section. Also, we added literature by Carrijo on ancestral enzymes in the main text (Line 70-79)
Comment 6: According to Ithentic, the work shows 23% similarity with literature; this rate needs to be reduced to 15% at most.
Answer: We performed a similarity check and found that the similarity for the current revised manuscript is 11 % excluding references. 23% of similarity checks come when the reference is included. 
Round 2
Reviewer 1 Report
Comments and Suggestions for Authors
All my concerns have been addressed and this manuscript is suitable for publication.